# Linear Time Computation of Moments in Sum-Product Networks

**Han Zhao**
Machine Learning Department
Carnegie Mellon University
Pittsburgh, PA 15213
han.zhao@cs.cmu.edu

**Geoff Gordon**
Machine Learning Department
Carnegie Mellon University
Pittsburgh, PA 15213
ggordon@cs.cmu.edu

## Abstract

Bayesian online algorithms for Sum-Product Networks (SPNs) need to update their posterior distribution after seeing one single additional instance. To do so, they must compute moments of the model parameters under this distribution. The best existing method for computing such moments scales quadratically in the size of the SPN, although it scales linearly for trees. This unfortunate scaling makes Bayesian online algorithms prohibitively expensive, except for small or tree-structured SPNs. We propose an optimal linear-time algorithm that works even when the SPN is a general directed acyclic graph (DAG), which significantly broadens the applicability of Bayesian online algorithms for SPNs. There are three key ingredients in the design and analysis of our algorithm: 1). For each edge in the graph, we construct a linear time reduction from the moment computation problem to a joint inference problem in SPNs. 2). Using the property that each SPN computes a multilinear polynomial, we give an efficient procedure for polynomial evaluation by differentiation without expanding the network that may contain exponentially many monomials. 3). We propose a dynamic programming method to further reduce the computation of the moments of all the edges in the graph from quadratic to linear. We demonstrate the usefulness of our linear time algorithm by applying it to develop a linear time assume density filter (ADF) for SPNs.

## 1 Introduction

Sum-Product Networks (SPNs) have recently attracted some interest because of their flexibility in modeling complex distributions as well as the tractability of performing exact marginal inference [11, 5, 6, 9, 16–18, 10]. They are general-purpose inference machines over which one can perform exact joint, marginal and conditional queries in linear time in the size of the network. It has been shown that discrete SPNs are equivalent to arithmetic circuits (ACs) [3, 8] in the sense that one can transform each SPN into an equivalent AC and vice versa in linear time and space with respect to the network size [13]. SPNs are also closely connected to probabilistic graphical models: by interpreting each sum node in the network as a hidden variable and each product node as a rule encoding context-specific conditional independence [1], every SPN can be equivalently converted into a Bayesian network where compact data structures are used to represent the local probability distributions [16]. This relationship characterizes the probabilistic semantics encoded by the network structure and allows practitioners to design principled and efficient parameter learning algorithms for SPNs [17, 18].

Most existing batch learning algorithms for SPNs can be straightforwardly adapted to the online setting, where the network updates its parameters after it receives one instance at each time step. This online learning setting makes SPNs more widely applicable in various real-world scenarios. This includes the case where either the data set is too large to store at once, or the network needs to adapt to the change of external data distributions. Recently Rashwan et al. [12] proposed an

online Bayesian Moment Matching (BMM) algorithm to learn the probability distribution of the model parameters of SPNs based on the method of moments. Later Jaini et al. [7] extended this algorithm to the continuous case where the leaf nodes in the network are assumed to be Gaussian distributions. At a high level BMM can be understood as an instance of the general assumed density filtering framework [14] where the algorithm finds an approximate posterior distribution within a tractable family of distributions by the method of moments. Specifically, BMM for SPNs works by matching the first and second order moments of the approximate tractable posterior distribution to the exact but intractable posterior. An essential sub-routine of the above two algorithms [12, 7] is to efficiently compute the exact first and second order moments of the one-step update posterior distribution (cf. 3.2). Rashwan et al. [12] designed a recursive algorithm to achieve this goal in linear time when the underlying network structure is a tree, and this algorithm is also used by Jaini et al. [7] in the continuous case. However, the algorithm only works when the underlying network structure is a tree, and a naive computation of such moments in a DAG will scale quadratically w.r.t. the network size. Often this quadratic computation is prohibitively expensive even for SPNs with moderate sizes.

In this paper we propose a linear time (and space) algorithm that is able to compute any moments of all the network parameters simultaneously even when the underlying network structure is a DAG. There are three key ingredients in the design and analysis of our algorithm: 1). A linear time reduction from the moment computation problem to the joint inference problem in SPNs, 2). A succinct evaluation procedure of polynomial by differentiation without expanding it, and 3). A dynamic programming method to further reduce the quadratic computation to linear. The differential approach [3] used for polynomial evaluation can also be applied for exact inference in Bayesian networks. This technique has also been implicitly used in the recent development of a concave-convex procedure (CCCP) for optimizing the weights of SPNs [18]. Essentially, by reducing the moment computation problem to a joint inference problem in SPNs, we are able to exploit the fact that the network polynomial of an SPN computes a multilinear function in the model parameters, so we can efficiently evaluate this polynomial by differentiation even if the polynomial may contain exponentially many monomials, provided that the polynomial admits a tractable circuit complexity. Dynamic programming can be further used to trade off a constant factor in space complexity (using two additional copies of the network) to reduce the quadratic time complexity to linear so that all the edge moments can be computed simultaneously in two passes of the network. To demonstrate the usefulness of our linear time sub-routine for computing moments, we apply it to design an efficient assumed density filter [14] to learn the parameters of SPNs in an online fashion. ADF runs in linear time and space due to our efficient sub-routine. As an additional contribution, we also show that ADF and BMM can both be understood under a general framework of moment matching, where the only difference lies in the moments chosen to be matched and how to match the chosen moments.

## 2 Preliminaries

We use $[n]$ to abbreviate $\{1, 2, \ldots, n\}$, and we reserve $\mathcal{S}$ to represent an SPN, and use $|\mathcal{S}|$ to mean the size of an SPN, i.e., the number of edges plus the number of nodes in the graph.

### 2.1 Sum-Product Networks

A sum-product network $\mathcal{S}$ is a computational circuit over a set of random variables $\mathbf{X} = \{X_1, \ldots, X_n\}$. It is a rooted directed acyclic graph. The internal nodes of $\mathcal{S}$ are sums or products and the leaves are univariate distributions over $X_i$. In its simplest form, the leaves of $\mathcal{S}$ are indicator variables $\mathbb{I}_{X=x}$, which can also be understood as categorical distributions whose entire probability mass is on a single value. Edges from sum nodes are parameterized with positive weights. Sum node computes a weighted sum of its children and product node computes the product of its children. If we interpret each node in an SPN as a function of leaf nodes, then the *scope* of a node in SPN is defined as the set of variables that appear in this function. More formally, for any node $v$ in an SPN, if $v$ is a terminal node, say, an indicator variable over $X$, then $\text{scope}(v) = \{X\}$, else $\text{scope}(v) = \cup_{\tilde{v} \in \text{Ch}(v)} \text{scope}(\tilde{v})$. An SPN is *complete* iff each sum node has children with the same scope, and is *decomposable* iff for every product node $v$, $\text{scope}(v_i) \cap \text{scope}(v_j) = \varnothing$ for every pair $(v_i, v_j)$ of children of $v$. It has been shown that every valid SPN can be converted into a complete and decomposable SPN with at most a quadratic increase in size [16] without changing the underlying distribution. As a result, in this work we assume that all the SPNs we discuss are complete and decomposable.

Let $\mathbf{x}$ be an instantiation of the random vector $\mathbf{X}$. We associate an unnormalized probability $V_k(\mathbf{x}; \mathbf{w})$ with each node $k$ when the input to the network is $\mathbf{x}$ with network weights set to be $\mathbf{w}$:

$$V_k(\mathbf{x}; \mathbf{w}) = \begin{cases} p(X_i = \mathbf{x}_i) & \text{if } k \text{ is a leaf node over } X_i \\ \prod_{j \in \text{Ch}(k)} V_j(\mathbf{x}; \mathbf{w}) & \text{if } k \text{ is a product node} \\ \sum_{j \in \text{Ch}(k)} w_{k,j} V_j(\mathbf{x}; \mathbf{w}) & \text{if } k \text{ is a sum node} \end{cases} \tag{1}$$

where $\text{Ch}(k)$ is the child list of node $k$ in the graph and $w_{k,j}$ is the edge weight associated with sum node $k$ and its child node $j$. The probability of a joint assignment $\mathbf{X} = \mathbf{x}$ is computed by the value at the root of $\mathcal{S}$ with input $\mathbf{x}$ divided by a normalization constant $V_{\text{root}}(\mathbf{1}; \mathbf{w})$: $p(\mathbf{x}) = V_{\text{root}}(\mathbf{x}; \mathbf{w})/V_{\text{root}}(\mathbf{1}; \mathbf{w})$, where $V_{\text{root}}(\mathbf{1}; \mathbf{w})$ is the value of the root node when all the values of leaf nodes are set to be 1. This essentially corresponds to marginalizing out the random vector $\mathbf{X}$, which will ensure $p(\mathbf{x})$ defines a proper probability distribution. Remarkably, all queries w.r.t. $\mathbf{x}$, including joint, marginal, and conditional, can be answered in linear time in the size of the network.

## 2.2 Bayesian Networks and Mixture Models

We provide two alternative interpretations of SPNs that will be useful later to design our linear time moment computation algorithm. The first one relates SPNs with Bayesian networks (BNs). Informally, any complete and decomposable SPN $\mathcal{S}$ over $\mathbf{X} = \{X_1, \ldots, X_n\}$ can be converted into a bipartite BN with $O(n|\mathcal{S}|)$ size [16]. In this construction, each internal sum node in $\mathcal{S}$ corresponds to one latent variable in the constructed BN, and each leaf distribution node corresponds to one observable variable in the BN. Furthermore, the constructed BN will be a simple bipartite graph with one layer of local latent variables pointing to one layer of observable variables $\mathbf{X}$. An observable variable is a child of a local latent variable if and only if the observable variable appears as a descendant of the latent variable (sum node) in the original SPN. This means that the SPN $\mathcal{S}$ can be understood as a BN where the number of latent variables per instance is $O(|\mathcal{S}|)$.

The second perspective is to view an SPN $\mathcal{S}$ as a mixture model with exponentially many mixture components [4, 18]. More specifically, we can decompose each complete and decomposable SPN $\mathcal{S}$ into a sum of induced trees, where each tree corresponds to a product of univariate distributions. To proceed, we first formally define what we called *induced trees*:

**Definition 1** (Induced tree SPN). Given a complete and decomposable SPN $\mathcal{S}$ over $\mathbf{X} = \{X_1, \ldots, X_n\}$, $\mathcal{T} = (\mathcal{T}_V, \mathcal{T}_E)$ is called an *induced tree SPN* from $\mathcal{S}$ if 1). $\text{Root}(\mathcal{S}) \in \mathcal{T}_V$; 2). If $v \in \mathcal{T}_V$ is a sum node, then exactly one child of $v$ in $\mathcal{S}$ is in $\mathcal{T}_V$, and the corresponding edge is in $\mathcal{T}_E$; 3). If $v \in \mathcal{T}_V$ is a product node, then all the children of $v$ in $\mathcal{S}$ are in $\mathcal{T}_V$, and the corresponding edges are in $\mathcal{T}_E$.

It has been shown that Def. 1 produces subgraphs of $\mathcal{S}$ that are trees as long as the original SPN $\mathcal{S}$ is complete and decomposable [4, 18]. One useful result based on the concept of induced trees is:

**Theorem 1** ([18]). Let $\tau_{\mathcal{S}} = V_{\text{root}}(\mathbf{1}; \mathbf{1})$. $\tau_{\mathcal{S}}$ counts the number of unique induced trees in $\mathcal{S}$, and $V_{\text{root}}(\mathbf{x}; \mathbf{w})$ can be written as $\sum_{t=1}^{\tau_{\mathcal{S}}} \prod_{(k,j) \in \mathcal{T}_{tE}} w_{k,j} \prod_{i=1}^{n} p_t(X_i = \mathbf{x}_i)$, where $\mathcal{T}_t$ is the $t$th unique induced tree of $\mathcal{S}$ and $p_t(X_i)$ is a univariate distribution over $X_i$ in $\mathcal{T}_t$ as a leaf node.

Thm. 1 shows that $\tau_{\mathcal{S}} = V_{\text{root}}(\mathbf{1}; \mathbf{1})$ can also be computed efficiently by setting all the edge weights to be 1. In general counting problems are in the #P complexity class [15], and the fact that both probabilistic inference and counting problem are tractable in SPNs also implies that SPNs work on subsets of distributions that have succinct/efficient circuit representation. Without loss of generality assuming that sum layers alternate with product layers in $\mathcal{S}$, we have $\tau_{\mathcal{S}} = \Omega(2^{H(\mathcal{S})})$, where $H(\mathcal{S})$ is the height of $\mathcal{S}$. Hence the mixture model represented by $\mathcal{S}$ has number of mixture components that is exponential in the height of $\mathcal{S}$. Thm. 1 characterizes both the number of components and the form of each component in the mixture model, as well as their mixture weights. For the convenience of later discussion, we call $V_{\text{root}}(\mathbf{x}; \mathbf{w})$ the *network polynomial* of $\mathcal{S}$.

**Corollary 1.** The *network polynomial* $V_{\text{root}}(\mathbf{x}; \mathbf{w})$ is a multilinear function of $\mathbf{w}$ with positive coefficients on each monomial.

Corollary 1 holds since each monomial corresponds to an induced tree and each edge appears at most once in the tree. This property will be crucial and useful in our derivation of a linear time algorithm for moment computation in SPNs.

## 3 Linear Time Exact Moment Computation

### 3.1 Exact Posterior Has Exponentially Many Modes

Let $m$ be the number of sum nodes in $\mathcal{S}$. Suppose we are given a fully factorized prior distribution $p_0(\mathbf{w}; \boldsymbol{\alpha}) = \prod_{k=1}^{m} p_0(w_k; \alpha_k)$ over $\mathbf{w}$. It is worth pointing out the fully factorized prior distribution is well justified by the bipartite graph structure of the equivalent BN we introduced in section 2.2. We are interested in computing the moments of the posterior distribution after we receive one observation from the world. Essentially, this is the Bayesian online learning setting where we update the belief about the distribution of model parameters as we observe data from the world sequentially. Note that $w_k$ corresponds to the weight vector associated with sum node $k$, so $w_k$ is a vector that satisfies $w_k > 0$ and $\mathbf{1}^T w_k = 1$. Let us assume that the prior distribution for each $w_k$ is Dirichlet, i.e.,

$$p_0(\mathbf{w}; \boldsymbol{\alpha}) = \prod_{k=1}^{m} \text{Dir}(w_k; \alpha_k) = \prod_{k=1}^{m} \frac{\Gamma(\sum_j \alpha_{k,j})}{\prod_j \Gamma(\alpha_{k,j})} \prod_j w_{k,j}^{\alpha_{k,j}-1}$$

After observing one instance $\mathbf{x}$, we have the exact posterior distribution to be: $p(\mathbf{w} \mid \mathbf{x}) = p_0(\mathbf{w}; \boldsymbol{\alpha}) p(\mathbf{x} \mid \mathbf{w}) / p(\mathbf{x})$. Let $Z_{\mathbf{x}} \triangleq p(\mathbf{x})$ and realize that the network polynomial also computes the likelihood $p(\mathbf{x} \mid \mathbf{w})$. Plugging the expression for the prior distribution as well as the network polynomial into the above Bayes formula, we have

$$p(\mathbf{w} \mid \mathbf{x}) = \frac{1}{Z_{\mathbf{x}}} \sum_{t=1}^{\tau_{\mathcal{S}}} \prod_{k=1}^{m} \text{Dir}(w_k; \alpha_k) \prod_{(k,j) \in \mathcal{T}_{tE}} w_{k,j} \prod_{i=1}^{n} p_t(x_i)$$

Since Dirichlet is a conjugate distribution to the multinomial, each term in the summation is an updated Dirichlet with a multiplicative constant. So, the above equation suggests that the exact posterior distribution becomes a mixture of $\tau_{\mathcal{S}}$ Dirichlets after one observation. In a data set of $D$ instances, the exact posterior will become a mixture of $\tau_{\mathcal{S}}^D$ components, which is intractable to maintain since $\tau_{\mathcal{S}} = \Omega(2^{H(\mathcal{S})})$.

The hardness of maintaining the exact posterior distribution appeals for an approximate scheme where we can sequentially update our belief about the distribution while at the same time efficiently maintain the approximation. Assumed density filtering [14] is such a framework: the algorithm chooses an approximate distribution from a tractable family of distributions after observing each instance. A typical choice is to match the moments of an approximation to the exact posterior.

### 3.2 The Hardness of Computing Moments

In order to find an approximate distribution to match the moments of the exact posterior, we need to be able to compute those moments under the exact posterior. This is not a problem for traditional mixture models including mixture of Gaussians, latent Dirichlet allocation, etc., since the number of mixture components in those models are assumed to be small constants. However, this is not the case for SPNs, where the effective number of mixture components is $\tau_{\mathcal{S}} = \Omega(2^{H(\mathcal{S})})$, which also depends on the input network $\mathcal{S}$.

To simplify the notation, for each $t \in [\tau_{\mathcal{S}}]$, we define $c_t \triangleq \prod_{i=1}^{n} p_t(x_i)^1$ and $u_t \triangleq \int_{\mathbf{w}} p_0(\mathbf{w}) \prod_{(k,j) \in \mathcal{T}_{tE}} w_{k,j} \, d\mathbf{w}$. That is, $c_t$ corresponds to the product of leaf distributions in the $t$th induced tree $\mathcal{T}_t$, and $u_t$ is the moment of $\prod_{(k,j) \in \mathcal{T}_{tE}} w_{k,j}$, i.e., the product of tree edges, under the prior distribution $p_0(\mathbf{w})$. Realizing that the posterior distribution needs to satisfy the normalization constraint, we have:

$$\sum_{t=1}^{\tau_{\mathcal{S}}} c_t \int_{\mathbf{w}} p_0(\mathbf{w}) \prod_{(k,j) \in \mathcal{T}_{tE}} w_{k,j} \, d\mathbf{w} = \sum_{t=1}^{\tau_{\mathcal{S}}} c_t u_t = Z_{\mathbf{x}} \tag{2}$$

Note that the prior distribution for a sum node is a Dirichlet distribution. In this case we can compute a closed form expression for $u_t$ as:

$$u_t = \prod_{(k,j) \in \mathcal{T}_{tE}} \int_{w_k} p_0(w_k) w_{k,j} \, dw_k = \prod_{(k,j) \in \mathcal{T}_{tE}} \mathbb{E}_{p_0(w_k)}[w_{k,j}] = \prod_{(k,j) \in \mathcal{T}_{tE}} \frac{\alpha_{k,j}}{\sum_{j'} \alpha_{k,j'}} \tag{3}$$

More generally, let $f(\cdot)$ be a function applied to each edge weight in an SPN. We use the notation $M_p(f)$ to mean the moment of function $f$ evaluated under distribution $p$. We are interested in computing $M_p(f)$ where $p = p(\mathbf{w} \mid \mathbf{x})$, which we call the *one-step update posterior distribution*. More specifically, for each edge weight $w_{k,j}$, we would like to compute the following quantity:

$$M_p(f(w_{k,j})) = \int_{\mathbf{w}} f(w_{k,j}) p(\mathbf{w} \mid \mathbf{x}) \, d\mathbf{w} = \frac{1}{Z_{\mathbf{x}}} \sum_{t=1}^{\tau_S} c_t \int_{\mathbf{w}} p_0(\mathbf{w}) f(w_{k,j}) \prod_{(k',j') \in \mathcal{T}_{tE}} w_{k',j'} \, d\mathbf{w} \quad (4)$$

We note that (4) is not trivial to compute as it involves $\tau_S = \Omega(2^{H(S)})$ terms. Furthermore, in order to conduct moment matching, we need to compute the above moment for each edge $(k, j)$ from a sum node. A naive computation will lead to a total time complexity $\Omega(|S| \cdot 2^{H(S)})$. A linear time algorithm to compute these moments has been designed by Rashwan et al. [12] when the underlying structure of $S$ is a tree. This algorithm recursively computes the moments in a top-down fashion along the tree. However, this algorithm breaks down when the graph is a DAG.

In what follows we will present a $O(|S|)$ time and space algorithm that is able to compute all the moments simultaneously for general SPNs with DAG structures. We will first show a linear time reduction from the moment computation in (4) to a joint inference problem in $S$, and then proceed to use the differential trick to efficiently compute (4) for each edge in the graph. The final component will be a dynamic program to simultaneously compute (4) for all edges $w_{k,j}$ in the graph by trading constant factors of space complexity to reduce time complexity.

### 3.3 Linear Time Reduction from Moment Computation to Joint Inference

Let us first compute (4) for a fixed edge $(k, j)$. Our strategy is to partition all the induced trees based on whether they contain the tree edge $(k, j)$ or not. Define $\mathcal{T}_F = \{\mathcal{T}_t \mid (k, j) \notin \mathcal{T}_t, t \in [\tau_S]\}$ and $\mathcal{T}_T = \{\mathcal{T}_t \mid (k, j) \in \mathcal{T}_t, t \in [\tau_S]\}$. In other words, $\mathcal{T}_F$ corresponds to the set of trees that do not contain edge $(k, j)$ and $\mathcal{T}_T$ corresponds to the set of trees that contain edge $(k, j)$. Then,

$$M_p(f(w_{k,j})) = \frac{1}{Z_{\mathbf{x}}} \sum_{\mathcal{T}_t \in \mathcal{T}_T} c_t \int_{\mathbf{w}} p_0(\mathbf{w}) f(w_{k,j}) \prod_{(k',j') \in \mathcal{T}_{tE}} w_{k',j'} \, d\mathbf{w}$$
$$+ \frac{1}{Z_{\mathbf{x}}} \sum_{\mathcal{T}_t \in \mathcal{T}_F} c_t \int_{\mathbf{w}} p_0(\mathbf{w}) f(w_{k,j}) \prod_{(k',j') \in \mathcal{T}_{tE}} w_{k',j'} \, d\mathbf{w} \quad (5)$$

For the induced trees that contain edge $(k, j)$, we have

$$\frac{1}{Z_{\mathbf{x}}} \sum_{\mathcal{T}_t \in \mathcal{T}_T} c_t \int_{\mathbf{w}} p_0(\mathbf{w}) f(w_{k,j}) \prod_{(k',j') \in \mathcal{T}_{tE}} w_{k',j'} \, d\mathbf{w} = \frac{1}{Z_{\mathbf{x}}} \sum_{\mathcal{T}_t \in \mathcal{T}_T} c_t u_t M_{p'_{0,k}}(f(w_{k,j})) \quad (6)$$

where $p'_{0,k}$ is the one-step update posterior Dirichlet distribution for sum node $k$ after absorbing the term $w_{k,j}$. Similarly, for the induced trees that do not contain the edge $(k, j)$:

$$\frac{1}{Z_{\mathbf{x}}} \sum_{\mathcal{T}_t \in \mathcal{T}_F} c_t \int_{\mathbf{w}} p_0(\mathbf{w}) f(w_{k,j}) \prod_{(k',j') \in \mathcal{T}_{tE}} w_{k',j'} \, d\mathbf{w} = \frac{1}{Z_{\mathbf{x}}} \sum_{\mathcal{T}_t \in \mathcal{T}_F} c_t u_t M_{p_{0,k}}(f(w_{k,j})) \quad (7)$$

where $p_{0,k}$ is the prior Dirichlet distribution for sum node $k$. The above equation holds by changing the order of integration and realize that since $(k, j)$ is not in tree $\mathcal{T}_t$, $\prod_{(k',j') \in \mathcal{T}_{tE}} w_{k',j'}$ does not contain the term $w_{k,j}$. Note that both $M_{p_{0,k}}(f(w_{k,j}))$ and $M_{p'_{0,k}}(f(w_{k,j}))$ are independent of specific induced trees, so we can combine the above two parts to express $M_p(f(w_{k,j}))$ as:

$$M_p(f(w_{k,j})) = \left( \frac{1}{Z_{\mathbf{x}}} \sum_{\mathcal{T}_t \in \mathcal{T}_F} c_t u_t \right) M_{p_{0,k}}(f(w_{k,j})) + \left( \frac{1}{Z_{\mathbf{x}}} \sum_{\mathcal{T}_t \in \mathcal{T}_T} c_t u_t \right) M_{p'_{0,k}}(f(w_{k,j})) \quad (8)$$

From (2) we have

$$\frac{1}{Z_{\mathbf{x}}} \sum_{t=1}^{\tau_S} c_t u_t = 1 \quad \text{and} \quad \sum_{t=1}^{\tau_S} c_t u_t = \sum_{\mathcal{T}_t \in \mathcal{T}_T} c_t u_t + \sum_{\mathcal{T}_t \in \mathcal{T}_F} c_t u_t$$

This implies that $M_p(f)$ is in fact a convex combination of $M_{p_{0,k}}(f)$ and $M_{p'_{0,k}}(f)$. In other words, since both $M_{p_{0,k}}(f)$ and $M_{p'_{0,k}}(f)$ can be computed in closed form for each edge $(k, j)$, so in order to compute (4), we only need to be able to compute the two coefficients efficiently. Recall that for each induced tree $\mathcal{T}_t$, we have the expression of $u_t$ as $u_t = \prod_{(k,j)\in\mathcal{T}_{tE}} \alpha_{k,j} / \sum_{j'} \alpha_{k,j'}$. So the term $\sum_{t=1}^{\tau_\mathcal{S}} c_t u_t$ can thus be expressed as:

$$\sum_{t=1}^{\tau_\mathcal{S}} c_t u_t = \sum_{t=1}^{\tau_\mathcal{S}} \prod_{(k,j)\in\mathcal{T}_{tE}} \frac{\alpha_{k,j}}{\sum_{j'} \alpha_{k,j'}} \prod_{i=1}^{n} p_t(x_i) \tag{9}$$

The key observation that allows us to find the linear time reduction lies in the fact that (9) shares exactly the same functional form as the network polynomial, with the only difference being the specification of edge weights in the network. The following lemma formalizes our argument.

**Lemma 1.** $\sum_{t=1}^{\tau_\mathcal{S}} c_t u_t$ can be computed in $O(|\mathcal{S}|)$ time and space in a bottom-up evaluation of $\mathcal{S}$.

*Proof.* Compare the form of (9) to the network polynomial:

$$p(\mathbf{x} \mid \mathbf{w}) = V_{\text{root}}(\mathbf{x}; \mathbf{w}) = \sum_{t=1}^{\tau_\mathcal{S}} \prod_{(k,j)\in\mathcal{T}_{tE}} w_{k,j} \prod_{i=1}^{n} p_t(x_i) \tag{10}$$

Clearly (9) and (10) share the same functional form and the only difference lies in that the edge weight used in (9) is given by $\alpha_{k,j} / \sum_{j'} \alpha_{k,j'}$ while the edge weight used in (10) is given by $w_{k,j}$, both of which are constrained to be positive and locally normalized. This means that in order to compute the value of (9), we can replace all the edge weights $w_{k,j}$ with $\alpha_{k,j} / \sum_{j'} \alpha_{k,j'}$, and a bottom-up pass evaluation of $\mathcal{S}$ will give us the desired result at the root of the network. The linear time and space complexity then follows from the linear time and space inference complexity of SPNs. ∎

In other words, we reduce the original moment computation problem for edge $(k, j)$ to a joint inference problem in $\mathcal{S}$ with a set of weights determined by $\boldsymbol{\alpha}$.

### 3.4 Efficient Polynomial Evaluation by Differentiation

To evaluate (8), we also need to compute $\sum_{\mathcal{T}_t\in\mathcal{T}_T} c_t u_t$ efficiently, where the sum is over a subset of induced trees that contain edge $(k, j)$. Again, due to the exponential lower bound on the number of unique induced trees, a brute force computation is infeasible in the worst case. The key observation is that we can use the *differential trick* to solve this problem by realizing the fact that $Z_\mathbf{x} = \sum_{t=1}^{\tau_\mathcal{S}} c_t u_t$ is a multilinear function in $\alpha_{k,j} / \sum_{j'} \alpha_{k,j'}, \forall k, j$ and it has a tractable circuit representation since it shares the same network structure with $\mathcal{S}$.

**Lemma 2.** $\sum_{\mathcal{T}_t\in\mathcal{T}_T} c_t u_t = w_{k,j} (\partial \sum_{t=1}^{\tau_\mathcal{S}} c_t u_t / \partial w_{k,j})$, and it can be computed in $O(|\mathcal{S}|)$ time and space in a top-down differentiation of $\mathcal{S}$.

*Proof.* Define $w_{k,j} \triangleq \alpha_{k,j} / \sum_{j'} \alpha_{k,j'}$, then

$$\sum_{\mathcal{T}_t\in\mathcal{T}_T} c_t u_t = \sum_{\mathcal{T}_t\in\mathcal{T}_T} \prod_{(k',j')\in\mathcal{T}_{tE}} w_{k',j'} \prod_{i=1}^{n} p_t(x_i)$$

$$= w_{k,j} \sum_{\substack{\mathcal{T}_t\in\mathcal{T}_T \ (k',j')\in\mathcal{T}_{tE} \\ (k',j')\neq(k,j)}} \prod w_{k',j'} \prod_{i=1}^{n} p_t(x_i) + 0 \cdot \sum_{\mathcal{T}_t\in\mathcal{T}_F} c_t u_t$$

$$= w_{k,j} \left( \frac{\partial}{\partial w_{k,j}} \sum_{\mathcal{T}_t\in\mathcal{T}_T} c_t u_t + \frac{\partial}{\partial w_{k,j}} \sum_{\mathcal{T}_t\in\mathcal{T}_F} c_t u_t \right) = w_{k,j} \left( \frac{\partial}{\partial w_{k,j}} \sum_{t=1}^{\tau_\mathcal{S}} c_t u_t \right)$$

where the second equality is by Corollary 1 that the network polynomial is a multilinear function of $w_{k,j}$ and the third equality holds because $\mathcal{T}_F$ is the set of trees that do not contain $w_{k,j}$. The last equality follows by simple algebraic transformations. In summary, the above lemma holds because of the fact that differential operator applied to a multilinear function acts as a selector for all the

monomials containing a specific variable. Hence, $\sum_{\mathcal{T}_t \in \mathcal{T}_F} c_t u_t = \sum_{t=1}^{\tau_S} c_t u_t - \sum_{\mathcal{T}_t \in \mathcal{T}_T} c_t u_t$ can also be computed. To show the linear time and space complexity, recall that the differentiation w.r.t. $w_{k,j}$ can be efficiently computed by back-propagation in a top-down pass of $\mathcal{S}$ once we have computed $\sum_{t=1}^{\tau_S} c_t u_t$ in a bottom-up pass of $\mathcal{S}$. ∎

**Remark**. The fact that we can compute the differentiation w.r.t. $w_{k,j}$ using the original circuit without expanding it underlies many recent advances in the algorithmic design of SPNs. Zhao et al. [18, 17] used the above differential trick to design linear time collapsed variational algorithm and the concave-convex produce for parameter estimation in SPNs. A different but related approach, where the differential operator is taken w.r.t. input indicators, not model parameters, is applied in computing the marginal probability in Bayesian networks and junction trees [3, 8]. We finish this discussion by concluding that when the polynomial computed by the network is a multilinear function in terms of model parameters or input indicators (such as in SPNs), then the differential operator w.r.t. a variable can be used as an efficient way to compute the sum of the subset of monomials that contain the specific variable.

### 3.5 Dynamic Programming: from Quadratic to Linear

Define $D_k(\mathbf{x}; \mathbf{w}) = \partial V_{\text{root}}(\mathbf{x}; \mathbf{w})/\partial V_k(\mathbf{x}; \mathbf{w})$. Then the differentiation term $\partial \sum_{t=1}^{\tau_S} c_t u_t / \partial w_{k,j}$ in Lemma 2 can be computed via back-propagation in a top-down pass of the network as follows:

$$\frac{\partial \sum_{t=1}^{\tau_S} c_t u_t}{\partial w_{k,j}} = \frac{\partial V_{\text{root}}(\mathbf{x}; \mathbf{w})}{\partial V_k(\mathbf{x}; \mathbf{w})} \frac{\partial V_k(\mathbf{x}; \mathbf{w})}{\partial w_{k,j}} = D_k(\mathbf{x}; \mathbf{w}) V_j(\mathbf{x}; \mathbf{w}) \tag{11}$$

Let $\lambda_{k,j} = \left( w_{k,j} V_j(\mathbf{x}; \mathbf{w}) D_k(\mathbf{x}; \mathbf{w}) \right) / V_{\text{root}}(\mathbf{x}; \mathbf{w})$ and $f_{k,j} = f(w_{k,j})$, then the final formula for computing the moment of edge weight $w_{k,j}$ under the one-step update posterior $p$ is given by

$$M_p(f_{k,j}) = (1 - \lambda_{k,j}) M_{p_0}(f_{k,j}) + \lambda_{k,j} M_{p'_0}(f_{k,j}) \tag{12}$$

**Corollary 2.** For each edges $(k, j)$, (8) can be computed in $O(|\mathcal{S}|)$ time and space.

The corollary simply follows from Lemma 1 and Lemma 2 with the assumption that the moments under the prior has closed form solution. By definition, we also have $\lambda_{k,j} = \sum_{\mathcal{T}_t \in \mathcal{T}_T} c_t u_t / Z_{\mathbf{x}}$, hence $0 \le \lambda_{k,j} \le 1, \forall (k,j)$. This formula shows that $\lambda_{k,j}$ computes the ratio of all the induced trees that contain edge $(k, j)$ to the network. Roughly speaking, this measures how important the contribution of a specific edge is to the whole network polynomial. As a result, we can interpret (12) as follows: the more important the edge is, the more portion of the moment comes from the new observation. We visualize our moment computation method for a single edge $(k, j)$ in Fig. 1.

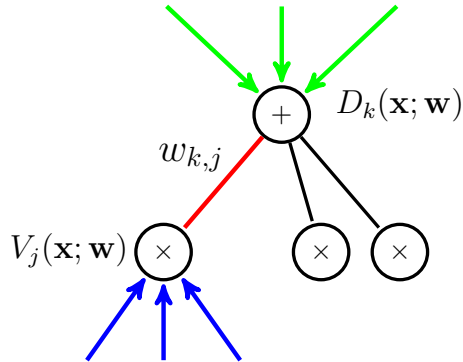

Figure 1: The moment computation only needs three quantities: the forward evaluation value at node $j$, the backward differentiation value node $k$, and the weight of edge $(k, j)$.

**Remark**. CCCP for SPNs was originally derived using a sequential convex relaxation technique, where in each iteration a concave surrogate function is constructed and optimized. The key update in each iteration of CCCP ([18], (7)) is given as follows: $w'_{k,j} \propto w_{k,j} V_j(\mathbf{x}; \mathbf{w}) D_k(\mathbf{x}; \mathbf{w})/V_{\text{root}}(\mathbf{x}; \mathbf{w})$, where the R.H.S. is exactly the same as $\lambda_{k,j}$ defined above. From this perspective, CCCP can also be understood as implicitly applying the differential trick to compute $\lambda_{k,j}$, i.e., the relative importance of edge $(k, j)$, and then take updates according to this importance measure.

In order to compute the moments of all the edge weights $w_{k,j}$, a naive computation would scale $O(|\mathcal{S}|^2)$ because there are $O(|\mathcal{S}|)$ edges in the graph and from Cor. 2 each such computation takes $O(|\mathcal{S}|)$ time. The key observation that allows us to further reduce the complexity to linear comes from the structure of $\lambda_{k,j}$: $\lambda_{k,j}$ only depends on three terms, i.e., the forward evaluation value

$V_j(\mathbf{x}; \mathbf{w})$, the backward differentiation value $D_k(\mathbf{x}; \mathbf{w})$ and the original weight of the edge $w_{k,j}$. This implies that we can use dynamic programming to cache both $V_j(\mathbf{x}; \mathbf{w})$ and $D_k(\mathbf{x}; \mathbf{w})$ in a bottom-up evaluation pass and a top-down differentiation pass, respectively. At a high level, we trade off a constant factor in space complexity (using two additional copies of the network) to reduce the quadratic time complexity to linear.

**Theorem 2.** For all edges $(k, j)$, (8) can be computed in $O(|\mathcal{S}|)$ time and space.

*Proof.* During the bottom-up evaluation pass, in order to compute the value $V_{\text{root}}(\mathbf{x}; \mathbf{w})$ at the root of $\mathcal{S}$, we will also obtain all the values $V_j(\mathbf{x}; \mathbf{w})$ at each node $j$ in the graph. So instead of discarding these intermediate $V_j(\mathbf{x}; \mathbf{w})$, we cache them by allocating additional space at each node $j$. So after one bottom-up evaluation pass of the network, we will also have all the $V_j(\mathbf{x}; \mathbf{w})$ for each node $j$, at the cost of one additional copy of the network. Similarly, during the top-down differentiation pass of the network, because of the chain rule, we will also obtain all the intermediate $D_k(\mathbf{x}; \mathbf{w})$ at each node $k$. Again, we cache them. Once we have both $V_j(\mathbf{x}; \mathbf{w})$ and $D_k(\mathbf{x}; \mathbf{w})$ for each edge $(k, j)$, from (12), we can get all the moments for all the weighted edges in $\mathcal{S}$ simultaneously. Because the whole process only requires one bottom-up evaluation pass and one top-down differentiation pass of $\mathcal{S}$, the time complexity is $2|\mathcal{S}|$. Since we use two additional copies of $\mathcal{S}$, the space complexity is $3|\mathcal{S}|$. $\blacksquare$

We summarize the linear time algorithm for moment computation in Alg. 1.

---

**Algorithm 1** Linear Time Exact Moment Computation

---

**Input:** Prior $p_0(\mathbf{w} \mid \boldsymbol{\alpha})$, moment $f$, SPN $\mathcal{S}$ and input $\mathbf{x}$.
**Output:** $M_p(f(w_{k,j})), \forall(k, j)$.
 1: $w_{k,j} \leftarrow \alpha_{k,j} / \sum_{j'} \alpha_{k,j'}, \forall(k, j)$.
 2: Compute $M_{p_0}(f(w_{k,j}))$ and $M_{p_0'}(f(w_{k,j})), \forall(k, j)$.
 3: Bottom-up evaluation pass of $\mathcal{S}$ with input $\mathbf{x}$. Record $V_k(\mathbf{x}; \mathbf{w})$ at each node $k$.
 4: Top-down differentiation pass of $\mathcal{S}$ with input $\mathbf{x}$. Record $D_k(\mathbf{x}; \mathbf{w})$ at each node $k$.
 5: Compute the exact moment for each $(k, j)$: $M_p(f_{k,j}) = (1 - \lambda_{k,j}) M_{p_0}(f_{k,j}) + \lambda_{k,j} M_{p_0'}(f_{k,j})$.

---

## 4 Applications in Online Moment Matching

In this section we use Alg. 1 as a sub-routine to develop a new Bayesian online learning algorithm for SPNs based on assumed density filtering [14]. To do so, we find an approximate distribution by minimizing the KL divergence between the one-step update posterior and the approximate distribution. Let $\mathcal{P} = \{q \mid q = \prod_{k=1}^{m} \text{Dir}(w_k; \beta_k)\}$, i.e., $\mathcal{P}$ is the space of product of Dirichlet densities that are decomposable over all the sum nodes in $\mathcal{S}$. Note that since $p_0(\mathbf{w}; \boldsymbol{\alpha})$ is fully decomposable, we have $p_0 \in \mathcal{P}$. One natural choice is to try to find an approximate distribution $q \in \mathcal{P}$ such that $q$ minimizes the KL-divergence between $p(\mathbf{w}|\mathbf{x})$ and $q$, i.e.,

$$\hat{p} = \underset{q \in \mathcal{P}}{\arg\min} \, \mathbb{KL}(p(\mathbf{w} \mid \mathbf{x}) \parallel q)$$

It is not hard to show that when $q$ is an exponential family distribution, which is the case in our setting, the minimization problem corresponds to solving the following moment matching equation:

$$\mathbb{E}_{p(\mathbf{w}|\mathbf{x})}[T(w_k)] = \mathbb{E}_{q(\mathbf{w})}[T(w_k)] \tag{13}$$

where $T(w_k)$ is the vector of sufficient statistics of $q(w_k)$. When $q(\cdot)$ is a Dirichlet, we have $T(w_k) = \log w_k$, where the log is understood to be taken elementwise. This principle of finding an approximate distribution is also known as *reverse information projection* in the literature of information theory [2]. As a comparison, information projection corresponds to minimizing $\mathbb{KL}(q \parallel p(\mathbf{w} \mid \mathbf{x}))$ within the same family of distributions $q \in \mathcal{P}$. By utilizing our efficient linear time algorithm for exact moment computation, we propose a Bayesian online learning algorithm for SPNs based on the above moment matching principle, called assumed density filtering (ADF). The pseudocode is shown in Alg. 2.

In the ADF algorithm, for each edge $w_{k,j}$ the above moment matching equation amounts to solving the following equation:

$$\psi(\beta_{k,j}) - \psi(\sum_{j'} \beta_{k,j'}) = \mathbb{E}_{p(\mathbf{w}|\mathbf{x})}[\log w_{k,j}]$$

where $\psi(\cdot)$ is the digamma function. This is a system of nonlinear equations about $\beta$ where the R.H.S. of the above equation can be computed using Alg. 1 in $O(|\mathcal{S}|)$ time for all the edges $(k, j)$. To efficiently solve it, we take $\exp(\cdot)$ at both sides of the equation and approximate the L.H.S. using the fact that $\exp(\psi(\beta_{k,j})) \approx \beta_{k,j} - \frac{1}{2}$ for $\beta_{k,j} > 1$. Expanding the R.H.S. of the above equation using the identity from (12), we have:

$$\exp\left(\psi(\beta_{k,j}) - \psi(\sum_{j'} \beta_{w,j'})\right) = \exp\left(\mathbb{E}_{p(\mathbf{w}|\mathbf{x})}[\log w_{k,j}]\right)$$

$$\Leftrightarrow \frac{\beta_{k,j} - \frac{1}{2}}{\sum_{j'} \beta_{k,j'} - \frac{1}{2}} = \left(\frac{\alpha_{k,j} - \frac{1}{2}}{\sum_{j'} \alpha_{k,j'} - \frac{1}{2}}\right)^{(1-\lambda_{k,j})} \times \left(\frac{\alpha_{k,j} + \frac{1}{2}}{\sum_{j'} \alpha_{k,j'} + \frac{1}{2}}\right)^{\lambda_{k,j}} \quad (14)$$

Note that $(\alpha_{k,j} - 0.5)/(\sum_{j'} \alpha_{k,j'} - 0.5)$ is approximately the mean of the prior Dirichlet under $p_0$ and $(\alpha_{k,j} + 0.5)/(\sum_{j'} \alpha_{k,j'} + 0.5)$ is approximately the mean of $p_0'$, where $p_0'$ is the posterior by adding one pseudo-count to $w_{k,j}$. So (14) is essentially finding a posterior with hyperparameter $\beta$ such that the posterior mean is approximately the weighted geometric mean of the means given by $p_0$ and $p_0'$, weighted by $\lambda_{k,j}$.

Instead of matching the moments given by the sufficient statistics, also known as the natural moments, BMM tries to find an approximate distribution $q$ by matching the first order moments, i.e., the mean of the prior and the one-step update posterior. Using the same notation, we want $q$ to match the following equation:

$$\mathbb{E}_{q(\mathbf{w})}[w_k] = \mathbb{E}_{p(\mathbf{w}|\mathbf{x})}[w_k] \quad \Leftrightarrow \quad \frac{\beta_{k,j}}{\sum_{j'} \beta_{k,j'}} = (1 - \lambda_{k,j})\frac{\alpha_{k,j}}{\sum_{j'} \alpha_{k,j'}} + \lambda_{k,j}\frac{\alpha_{k,j} + 1}{\sum_{j'} \alpha_{k,j'} + 1} \quad (15)$$

Again, we can interpret the above equation as to find the posterior hyperparameter $\beta$ such that the posterior mean is given by the weighted arithmetic mean of the means given by $p_0$ and $p_0'$, weighted by $\lambda_{k,j}$. Notice that due to the normalization constraint, we cannot solve for $\beta$ directly from the above equations, and in order to solve for $\beta$ we will need one more equation to be added into the system. However, from line 1 of Alg. 1, what we need in the next iteration of the algorithm is not $\beta$, but only its normalized version. So we can get rid of the additional equation and use (15) as the update formula directly in our algorithm.

Using Alg. 1 as a sub-routine, both ADF and BMM enjoy linear running time, sharing the same order of time complexity as CCCP. However, since CCCP directly optimizes over the data log-likelihood, in practice we observe that CCCP often outperforms ADF and BMM in log-likelihood scores.

---

**Algorithm 2** Assumed Density Filtering for SPN

---

**Input:** Prior $p_0(\mathbf{w} \mid \boldsymbol{\alpha})$, SPN $\mathcal{S}$ and input $\{\mathbf{x}_i\}_{i=1}^{\infty}$.
1: $p(\mathbf{w}) \leftarrow p_0(\mathbf{w} \mid \boldsymbol{\alpha})$
2: **for** $i = 1, \ldots, \infty$ **do**
3:     Apply Alg. 1 to compute $\mathbb{E}_{p(\mathbf{w}|\mathbf{x}_i)}[\log w_{k,j}]$ for all edges $(k, j)$.
4:     Find $\hat{p} = \arg\min_{q \in \mathcal{P}} \mathbb{KL}(p(\mathbf{w} \mid \mathbf{x}_i) \| q)$ by solving the moment matching equation (13).
5:     $p(\mathbf{w}) \leftarrow \hat{p}(\mathbf{w})$.
6: **end for**

---

## 5   Conclusion

We propose an optimal linear time algorithm to efficiently compute the moments of model parameters in SPNs under online settings. The key techniques used in the design of our algorithm include the liner time reduction from moment computation to joint inference, the differential trick that is able to efficiently evaluate a multilinear function, and the dynamic programming to further reduce redundant computations. Using the proposed algorithm as a sub-routine, we are able to improve the time complexity of BMM from quadratic to linear on general SPNs with DAG structures. We also use the proposed algorithm as a sub-routine to design a new online algorithm, ADF. As a future direction, we hope to apply the proposed moment computation algorithm in the design of efficient structure learning algorithms for SPNs. We also expect that the analysis techniques we develop might find other uses for learning SPNs.

## Acknowledgements

HZ thanks Pascal Poupart for providing insightful comments. HZ and GG are supported in part by ONR award N000141512365.

## Footnotes

[1]For ease of notation, we omit the explicit dependency of $c_t$ on the instance $\mathbf{x}$.

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
