[Reviews · NeurIPS 2017]

Reviewer 1



This paper proposes a method to compute moments in SPNs in linear time and space, which is a useful procedure to perform Bayesian updating using moment matching approaches. This is indeed an important topic which is worth investigating. It seems that the derivation is quite straightforward and the lack of a practical use puts this submission in a different position from the "usual" NIPS paper. My main concern is that I was not able to follow the result of Corollary 2 and Theorem 2, which are the most important of the paper. It is said to follow from Lemma 1 and 2, but the definitions of the trees T_F and T_T need different parameters to be encoded/propagated depending on the edge (k,j). I was not able to see how to avoid the quadratic complexity. On the same note, it is mentioned in the introduction that the task is known to be computable in quadratic time, but later (beginning of page 5) it feels that not even quadratic is known. I think the details of the algorithm and its complexity could be made clearer. Minor issues: - Equations are not punctuated. - Equations are referred without parenthesis enclosing the number. - Some acronyms are not defined (even if they are obvious, they should be defined). - Notation between V(.;.) and V(.|.) is incompatible, see Eq.(1). - |S| is defined as the size of the graph of the SPN (which is not necessarily the size of the SPN, even if proportional). - v_i and v_j (line 80) could be mentioned "...every pair (v_i,v_j) of children...". - After line 175, "(4) =" is not so visually appealing. Better to use M_p(f(w_{k,j})). - "So...thus" of lines 185-186 sounds repetitive. ===after authors' response=== The query about the theorem has been clarified to me.

Reviewer 2



This paper solves an important problem of computing moments in SPNs. The reader wonders whether this paper only helps with Dirichlet priors on sum node weights. How does this interact with priors on leaf nodes or other sum node weight priors? It is disappointing to not see any experiments, especially comparing against heuristic methods to smooth SPN weights with false counts. [Post-rebuttal: I disagree that heuristic methods of parameter estimation are not comparable to methods proposed by this paper. Readers would be very curious how Hard/Soft Gradient Descent or E.M. with false count smoothing (for which each step uses less space and time compared to the proposed method) compares in terms of space/time per step and overall measures of likelihood and rates of convergence (accuracy and speed).] Section 3 The remarks are somewhat verbose. Algorithm 1 should be self-contained and describe exact expressions ("based on Eq. 11" is a tad informal) Section 4 It would be best if Algorithm 2 was self-contained and did not require resorting to the appendix. Appendix Eqn. 16, 17 and l. 327: Should correct the ambiguous denominators to show what is inside the sum.

Reviewer 3



I thank the authors for their response. As mentioned in my review, I also see the contributions of this paper on an algorithmic level. Nevertheless I would encourage the authors to at least comment on the existing empirical results to give readers a more complete picture. -- The submitted manuscript develops a linear time algorithm for computation of moments in SPNs with general DAG structure, improving upon the quadratic time complexity of existing algorithms. I like the contributions of this paper and I think the proposed algorithms are novel (given that the authors are the same as those of an arXiv paper from a couple of months ago) and could turn out to be useful. On the other hand, the contributions of the paper are somewhat limited. It would be beneficial to demonstrate the usefulness of the algorithms on some task like parameter estimation. In particular, it would be interesting to see whether the uncertainties that can be computed from the approximate posterior are useful in some way, e.g. to quantify uncertainty in queries etc. Regarding "remark 3" in lines 228-236. Given that the authors draw this close connections, the neural networks with products unit studied in the past should be mentioned. These product units are more general than the products computed in SPNs (they have tuneable parameters). A couple of more comments: * Lines 251 to 254 give the impression that the stated result about moment matching in the case of exponential family distributions as approximating distributions is novel/from the authors. I found the same result in Shiliang Sun, "A review of deterministic approximate inference techniques for Bayesian machine learning", 2013 and expect there are even older similar results. * There seems to be an arXiv version of the same paper which I have seen some time ago that goes beyond the submitted paper, e.g. it includes experiments. In these experiments, the method proposed in this paper do "only" compare favourably to other methods in terms of runtime, but not in terms of model fit. That is ok, as I see the contribution of this paper on a different level. However, as these results are known, I feel that they should have been mentioned somewhere (could have been anonymous).